# Hypoxemia prediction in pediatric patients under general anesthesia using machine learning: A retrospective observational study and external validation

Sujin Baek[1,2☯], Jung-Bin Park[3☯], Jihye Heo[4], Kyungsang Kim[5], Donghyeon Baek[6], Chahyun Oh[1,2], Hyung-Chul Lee[3], Dongheon Lee[4,7‡*], Boohwi Hong[1,2,8‡*]

1 Department of Anesthesiology and Pain Medicine, Chungnam National University Hospital, Daejeon, Republic of Korea, 2 Department of Anesthesiology and Pain Medicine, Chungnam National University College of Medicine, Daejeon, Republic of Korea, 3 Department of Anesthesiology and Pain Medicine, Seoul National University Hospital, Seoul, Republic of Korea, 4 Department of Radiology, Seoul National University College of Medicine, Seoul National University Hospital, Seoul, Republic of Korea, 5 Department of Radiology, Massachusetts General Hospital and Harvard Medical School, Massachusetts, United States of America, 6 Chungnam National University College of Medicine, Daejeon, Republic of Korea, 7 Institute of Medical and Biological Engineering, Seoul National University Medical Research Center, Seoul, Republic of Korea, 8 Big Data Center, Department of Medical Information, Chungnam National University Hospital, Daejeon, Korea

☯ These authors contributed equally to this work.
‡ These authors also contributed equally to this work.
* koho0127@gmail.com (BH); dhlee.jubilee@gmail.com (DL)

## Abstract

### Background

Pediatric patients under general anesthesia are particularly vulnerable to hypoxemia, which can lead to rapid oxygen desaturation. This vulnerability necessitates heightened vigilance from anesthesiologists, making pediatric anesthesia management especially challenging. Continuous intraoperative monitoring of oxygenation is critical. However, traditional methods relying solely on $SpO_2$ readings may be insufficient and prone to inaccuracies.

### Methods

This study aimed to develop and externally validate various machine learning models to predict hypoxemia in pediatric patients under general anesthesia. This retrospective observational study included 800 pediatric cases from Seoul National University Hospital and 134 pediatric cases from Chungnam National University Hospital. Patient data, including vital signs and ventilator parameters sampled every 2 seconds, were analyzed. Four machine learning models (XGBoost, LSTM, InceptionTime, and Transformer) were evaluated using area under the receiver operating characteristic curve (AUROC), area under the precision-recall curve (AUPRC), and F1-score.

**Data availability statement:** The data supporting the findings of this study are not publicly available due to patient confidentiality and ethical restrictions on the use of medical data. The initial Institutional Review Board (IRB) approvals for this study limited data use to internal research and did not include provisions for public data deposition. However, de-identified data may be made available from the corresponding author upon reasonable request and with approval from the relevant institutional review boards. Contact Information for Data Access Requests: - Chungnam National University Hospital Institutional Review Board: cnuhirb@cnuh.co.kr, +82-42-280-6713 - Seoul National University Hospital Institutional Review Board: irb@snuh.org, +82-2-2072-0694.

**Funding:** "This research was supported and funded by the SNUH Lee Kun-hee Child Cancer & Rare Disease Project, Republic of Korea (grant number: 24C-003-0100) and supported by research fund from Chungnam National University."

## Results

XGBoost achieved the highest performance in internal validation (AUROC, 0.85), whereas the Transformer model demonstrated the best performance in external validation (AUROC, 0.83). Reducing the observation window from 1 minute to 10 seconds lowered the AUPRC but preserved high AUROC.

## Conclusions

The XGBoost and Transformer models demonstrated robust performance in predicting intraoperative hypoxemia in pediatric patients under general anesthesia across two hospitals. Adjustments for age-related variations did not enhance model performance. Future research should focus on developing machine learning models that can accurately distinguish true hypoxemia, leading to clinically significant improvements in patient outcomes.

## Introduction

Anesthesia-related cardiac arrest in pediatric patients is a significant concern, with approximately 27% of cases attributed to respiratory causes [1]. During surgical procedures, pediatric patients have an increased susceptibility to hypoxemia compared with adults, presenting a potentially fatal risk [2,3]. This heightened vulnerability stems from their smaller functional residual capacity, leading to reduced oxygen reserves and greater chest wall compliance, which predisposes them to small airway collapse [4,5]. Consequently, interruptions or inadequacies in oxygen supply can lead to a rapid decline in arterial oxygen levels. Therefore, vigilant intraoperative monitoring of oxygenation is imperative in pediatric patients.

However, false hypoxemia can complicate monitoring in pediatric patients, increase the workload of anesthesiologists, and obstruct careful monitoring efforts. A reliable study using Anesthesia Information Management Systems found that the incidence of pediatric intraoperative hypoxemia was 12%, with only 54% classified as true hypoxemia [2]. Conventionally, true hypoxemia is defined by a low arterial oxygen pressure ($PaO_2$) from arterial blood gas analysis. Relying solely on peripheral oxygen saturation ($SpO_2$) readings during hypoxemia events is insufficient for accurately assessing the condition of the patient. Thus, immediate assessment of hypoxemia requires evaluating various other patient-specific parameters. Additionally, parameters such as airway pressure and end-tidal carbon dioxide ($EtCO_2$) waveform, along with related metrics, influence intuitive interpretation during monitoring [6,7]. Therefore, clinicians rely on $SpO_2$ as a real-time surrogate, although it can be prone to artifacts.

Recent studies have reported the use of machine learning models, such as gradient boosting machine (GBM) and long short-term memory (LSTM), to predict hypoxemia in pediatric patients undergoing general anesthesia [8,9]. Lundberg et al. also achieved high predictive performance for hypoxemia, with an area under the receiver operating characteristic curve (AUROC) of up to 0.92, using machine learning-based

explainable models. However, their study was limited to internal validation, failing to confirm the generalizability of the models [10]. In addition, Erion et al. [11] and Liu et al. [12] demonstrated the effectiveness of $SpO_2$-based predictive models and proposed hybrid networks to address class imbalance and the challenge of predicting persistent hypoxemia. However, these models primarily relied on low-resolution data measured at 1-minute intervals and long observation windows (10 minutes to 1 hour), limiting their ability to capture the rapid desaturation events typical in pediatric hypoxemia, which occur within an average of 45 seconds [11,12]. Park et al. achieved high accuracy, recording an AUROC of up to 0.939, using high-frequency data sampled at 2-second intervals to improve temporal resolution. However, like previous studies, the lack of external validation limits the assessment of the model's generalizability across diverse clinical settings [13].

This study aimed to propose a machine learning model to predict intraoperative hypoxemia in pediatric patients undergoing general anesthesia. We used high-resolution time-series data normalized and stratified by age groups to account for pediatric-specific characteristics and compared the performance of various machine learning models. Additionally, external validation with data from other hospital was performed to verify model generalizability, and the importance of key features influencing predictions was analyzed.

## Materials and methods

### Study design

This retrospective multicenter external validation study was approved by the institutional review boards (IRBs) of Chungnam National University Hospital (CHUH; IRB number 2023-09-052) and Seoul National University Hospital (SNUH; IRB number H-2303-092-1412). Authorization for data anonymization and external transfer was granted by the Data Review Board of SNUH (DRB number DRB-R-2023-03-05).

This study used data from 800 pediatric cases at Seoul National University Hospital and 134 cases at Chungnam National University Hospital (Table 1). All biosignal data were obtained from the prospective registry of vital signs for surgical patients at SNUH and CNUH, using Vital Recorder version 1.9 (accessed at https://vitaldb.net; VitalDB, Seoul,

**Table 1. Baseline characteristics of the datasets.**

|  | SNUH (n = 800) | CNUH (n = 134) | p-Value |
|---|---|---|---|
| Age | 4.0 [8.25] | 5.0 [9.58] | 0.329 |
| 0–2 years | 315 (39.4) | 53 (39.6) | |
| 2–8 years | 352 (44.0) | 59 (44.0) | |
| 8–18 years | 131 (16.4) | 22 (16.4) | |
| Sex | | | |
| Male | 458 (57.3) | 86 (64.1) | 1 |
| Female | 342 (42.8) | 48 (35.8) | |
| Weight (kg) | 17.0 [26.9] | 18.3 [30.2] | 0.168 |
| Height (cm) | 103.1 [66.0] | 107.2 [73.3] | 0.585 |
| Hypoxemic event | 1,333 (1.67/case) | 168 (1.25/case) | <0.001 |
| 1/case | 494 (61.8) | 107 (79.9) | |
| 2/case | 151 (18.9) | 22 (16.4) | |
| 3/case | 83 (10.4) | 3 (2.2) | |
| 4/case | 72 (9.0) | 2 (1.5) | |
| Hypoxemic event length* (seconds) | 20.0 [32.0] | 34.5 [40.3] | <0.001 |

Data are presented as median [interquartile range] and number (%).

*Hypoxemic event length refers to the length of a single hypoxemic event.

SNUH, Seoul National University Hospital; CNUH, Chungnam National University Hospital.

Republic of Korea) [14]. The authors accessed the datasets for research purposes on 01/06/2023 at SNUH and on 15/12/2023 at CNUH. All data were de-identified, and no identifiable information was accessible to the authors. This manuscript adheres to the TRIPOD-AI (Transparent Reporting of a multivariable prediction model for Individual Prognosis Or Diagnosis-Artificial Intelligence) statement (S1 Table).

As described in Table 1, demographic data, including sex, age, height, and weight, were extracted from the electronic medical record. Surgical and anesthesia-related details were collected, including the surgery name, diagnosis, surgery date, and start and end times of surgery and anesthesia. The Vital Recorder program was used to gather physiological data, including vital signs (including $SpO_2$) and parameters from the anesthesia ventilator machine, including $EtCO_2$, fraction of inspired oxygen ($FiO_2$), tidal volume measured on exhalation (TV), peak inspiratory pressure (PIP), and minute ventilation (MV).

The inclusion criteria were pediatric patients younger than 18 years, those who underwent general anesthesia in the operating room, and those whose oxygen saturation decreased <95% at least once during surgery. Patients without Vital Recorder data during anesthesia, including patients undergoing cardiac or pulmonary surgery, those who underwent procedures that induce apnea (such as airway surgeries), and those with preoperative oxygen saturation <95% were excluded. This exclusion was intended to minimize confounding factors from procedure-specific causes of hypoxemia, thereby focusing the model on predicting events arising from general anesthetic management. Additionally, to exclude clinically insignificant and noisy hypoxemic events, the following exclusion criteria were applied: when the pulse rate measured via pulse oximetry exceeded 20% of the heart rate recorded by an electrocardiogram; when the plethysmography waveform was severely distorted; when an anesthesiologist recorded that the oxygen saturation was inaccurate (Fig 1).

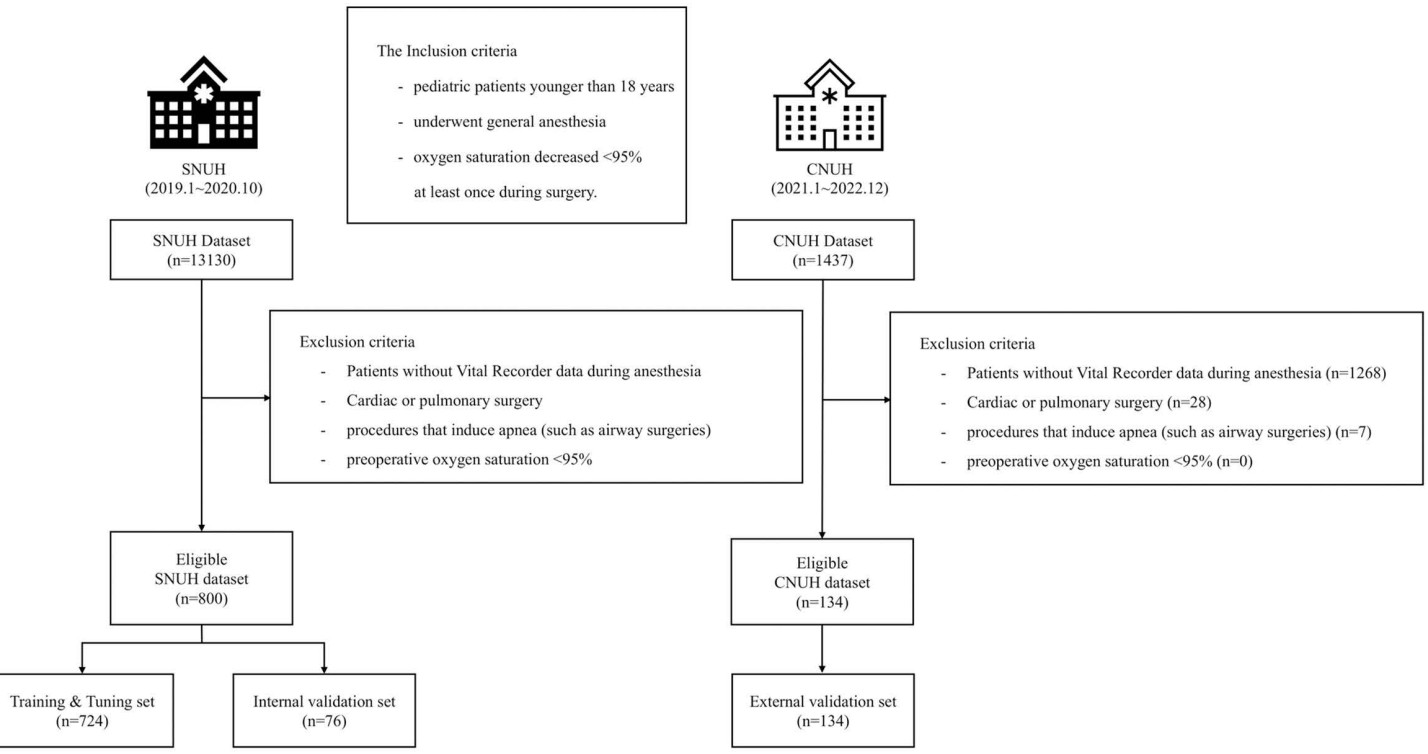

**Fig 1. Flowchart of patient inclusion.** SNUH, Seoul National University Hospital; CNUH, Chungnam National University Hospital.

## Preprocessing

A hypoxemic event was defined as the period from the onset of $SpO_2$ decline to 95%, through its nadir, and until $SpO_2$ values recovered to 95%. Clinically important time points in the vital sign data included the onset of $SpO_2$ decline, the point when $SpO_2$ dropped to 95%, the nadir of $SpO_2$, and the time when $SpO_2$ returned to baseline levels.

The biosignal data used in this study were sampled at 2-second intervals, with each time-series segment arranged into 120-second intervals. To predict potential hypoxemia events, the first half of each segment (30 time points consisting of 6 physiological parameters) was used as input for the model to forecast events occurring in the latter half of the segment.

Each segment for model training was set to 1 minute in length, with segments designated as hypoxemia-predictive intervals if a hypoxemia event occurred within 1 minute after the segment's end. This 1-minute prediction window was selected as it provides a clinically actionable timeframe for preventive interventions and is consistent with methodologies used in prior studies on pediatric hypoxemia prediction [13]. All other segments were labeled as non-hypoxemia intervals (Fig 2). Data segments of 1-minute duration were used for model training. These segments were generated with an initiation time staggered by 2 seconds from the preceding segment. To ensure that the $SpO_2$-defined labels reflected clinically valid hypoxemic events, all labeling processes were conducted by two expert anesthesiologists and were double-checked by inter-observers; any disagreements during the validation process were resolved through discussion.

The vital signs used for model input were sequential tabular data, comprising numerical time-series data sampled at 2-second intervals. The six key physiological parameters used were $SpO_2$, $FiO_2$, $EtCO_2$, PIP, TV, and MV. These time-series features were combined with four static demographic variables (height, weight, sex, and age) to form the complete input for the model. To prepare these features for the models, we first imputed any missing values with zero. Subsequently, for each 60-second input window, each of the six time-series signals was independently scaled to a range of [0, 1] using local min-max normalization to ensure stable model training.

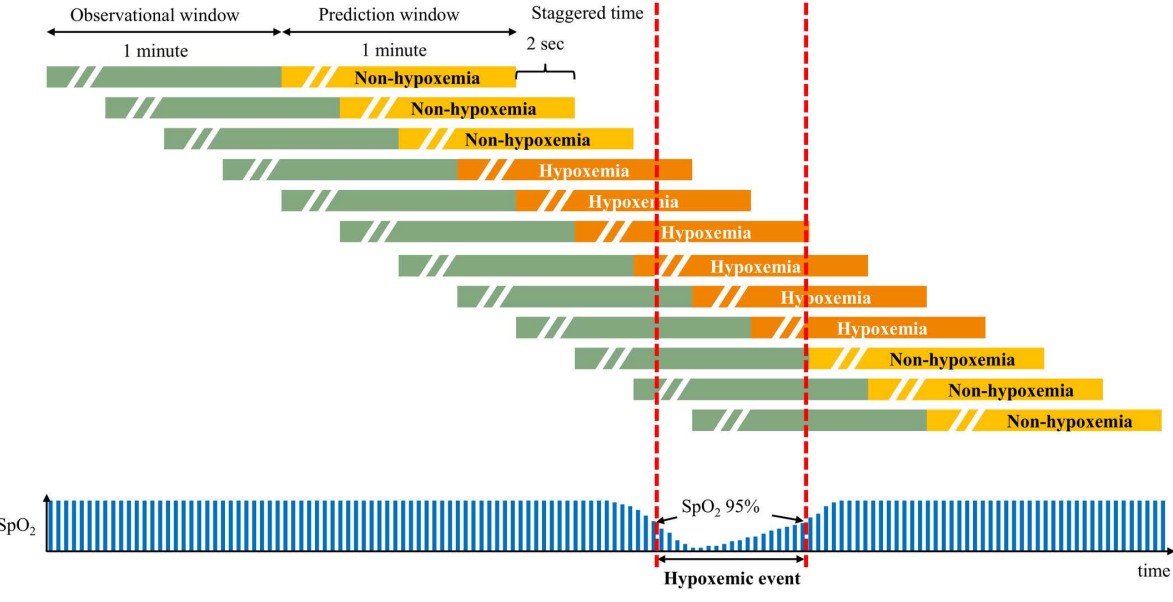

**Fig 2. Process of biosignal annotation for hypoxemia prediction.** Each 1-minute segment was labeled as hypoxemic if a hypoxemic event occurred within 1 minute after the segment ended and as non-hypoxemic otherwise. Yellow bar, prediction window with non-hypoxemic event; orange bar, prediction window including $SpO_2 < 95\%$ during hypoxemic event; green bar, observational window. $SpO_2$, peripheral oxygen saturation.

## Data preparation

Patients at SNUH from January 2019 to October 2020 were included, with the dataset split into training and internal validation sets at an 8:2 ratio, yielding 724 and 76 patients, respectively, based on surgery dates. The training set was further split into training and tuning sets at an 8:2 ratio for model training. For external validation, the dataset included 134 patients from CNUH between January 2021 and December 2022.

The proportion of data segments labeled as hypoxemia was higher in the CNUH dataset (1.42%) than in the SNUH dataset (1.15%). This difference reflects that hypoxemia events occurred more frequently but with shorter durations at SNUH, whereas they were less frequent but had longer durations at CNUH (S2 Table).

## Machine learning models and hyperparameter optimization

This study used four types of machine learning models: XGBoost, LSTM, InceptionTime, and Transformer [9,15–17]. To optimize each model, we performed a hyperparameter search for XGBoost, LSTM, and Transformer using a 5-fold cross-validation strategy on the training set. The parameter combination that yielded the highest average AUROC was selected as the final configuration. For the InceptionTime model, we adopted the architecture from the original publication due to its proven performance on time-series classification tasks. The complete search space and the final selected hyperparameters for all models are detailed in S3 Table.

We used XGBoost, which offers several advantages for biosignal analysis, including high computational efficiency, the ability to handle missing data, robustness to noise, and strong performance with structured and time-series data due to its gradient boosting framework [15,18]. We used 2,000 trees with a maximum depth of 5, a subsampling rate of 0.5, a gamma value of 0.4, and a minimum child weight of 2.

We also used LSTM, which is well-suited for processing time-series data and has been used in previous hypoxemia prediction studies [10–13], to effectively capture the long-term dependencies of biosignals [19]. Our LSTM model had a single hidden layer with 64 hidden nodes and 16 dense nodes along with a dropout rate of 0.5.

Additionally, InceptionTime is a scalable deep-learning model for time-series classification, inspired by the Inception-v4 architecture [16]. We used 8 filters as well as residual blocks and bottlenecks and had a depth of 6 and kernel size of 20.

Lastly, the Transformer has the ability to capture long-range dependencies through self-attention mechanisms, efficient parallel processing for large datasets, and adaptability to complex, multivariate time-series data [20]. The Transformer model featured 64 filters, 3 attention heads, an embedding dimension of 32, 1 convolutional layer, 3 transformer layers, and a dropout rate of 0.2.

## Implementation details

All models were trained for 100 epochs using five-fold cross-validation to improve robustness, with binary cross-entropy loss as the optimization criterion. As hypoxemic events were nearly 100 times rarer than non-hypoxemic periods, the loss function for class 1 was heavily weighted compared with class 0 to address the data imbalance issue.

The batch size was set to 1,024, and all deep learning models used the Adam optimizer with an initial learning rate of 0.001, beta1 of 0.9, and beta2 of 0.999. The model weights that achieved the best area under the receiver operating characteristic curve (AUROC) across all epochs were selected for evaluation. The F1 score was calculated, and the optimal threshold for binary classification was chosen based on the highest F1 score.

The models were developed on an NVIDIA GEFORCE RTX 4090 and implemented using Python version 3.10.1, TensorFlow version 2.10.1, and Keras version 2.10.0. For further details, the source code is publicly available on github at https://github.com/jihyeheo/ML-PredGA-Hypoxemia.

## Evaluation metrics and feature importance

The models' performance was evaluated using the AUROC, AUPRC, and F1 score. AUROC assesses the overall classification performance, AUPRC focuses on precision and recall in imbalanced datasets, and the F1 score provides a balance between precision and recall.

To further interpret the model's predictions and understand feature importance, we used Shapley Additive exPlanations (SHAP) values. The SHAP values, derived from cooperative game theory, provide a unified measure of feature importance by considering all possible combinations of feature values [21]. SHAP values are essential for understanding which features are most important to a model and how different values of these features influence the predictions.

## Statistical analysis

Statistical analyses were conducted using Python version 3.10.1. The normality of the data distribution was assessed using the Shapiro–Wilk test. Based on these results, continuous variables were reported as medians and interquartile ranges (IQRs). To evaluate data comparability, the Mann–Whitney U-test was used for continuous variables. Categorical variables were presented as numbers (%) and analyzed using Pearson's chi-square test. A comparison of the AUROCs between the models was conducted, and statistical significance was assessed using the DeLong test.

# Results

## Model performance for predicting hypoxemia

The models' performance in predicting intraoperative hypoxemia in pediatric patients under general anesthesia was evaluated using AUROC, AUPRC, and F1 score values across the four models in both the internal and external validation datasets (Table 2 and Fig 3). In the internal validation dataset, the XGBoost model achieved the highest performance among the four models, with an AUROC of 0.85, an AUPRC of 0.18, and an F1 score of 0.24, consistent with previous studies. However, it showed the lowest performance on the external dataset. Following the XGBoost, the InceptionTime, Transformer, and LSTM models were ranked in descending order of performance in the internal validation dataset. In the external validation dataset, the Transformer model demonstrated the best performance, with an AUROC of 0.85, AUPRC of 0.06, and F1 score of 0.12, followed by the LSTM and InceptionTime models in terms of effectiveness. Nonetheless, the AUPRC and F1 score values were notably low across all four models.

## Comparative performance in hypoxemia prediction

We evaluated the impact of four adjustment methods to improve model performance. First, we implemented data normalization to account for characteristics that vary significantly with age in pediatric patients. The performance of the

Table 2. Comparative performance of machine learning models for hypoxemia prediction in pediatric patients. The performance of the four machine learning models (XGBoost, LSTM, Transformer, and InceptionTime) in predicting hypoxemia in pediatric patients under general anesthesia was evaluated across both internal and external validation datasets.

| | Internal validation | | | | External validation | | | |
|---|---|---|---|---|---|---|---|---|
| | AUROC | AUPRC | F1 score | p-Value | AUROC | AUPRC | F1 score | p-Value |
| XGBoost | **0.8550** | **0.1816** | **0.2382** | | 0.7857 | 0.0402 | 0.0824 | <0.001 |
| LSTM | 0.7581 | 0.0536 | 0.1256 | <0.001 | 0.8488 | **0.0605** | **0.1347** | 0.3802 |
| Transformer | 0.7934 | 0.0505 | 0.1283 | <0.001 | **0.8501** | 0.0595 | 0.1227 | |
| InceptionTime | 0.8341 | 0.0990 | 0.1653 | <0.001 | 0.7974 | 0.0453 | 0.0954 | <0.001 |

LSTM, long short-term memory; AUROC, area under the receiver operating characteristic curve; AUPRC, area under the precision-recall curve

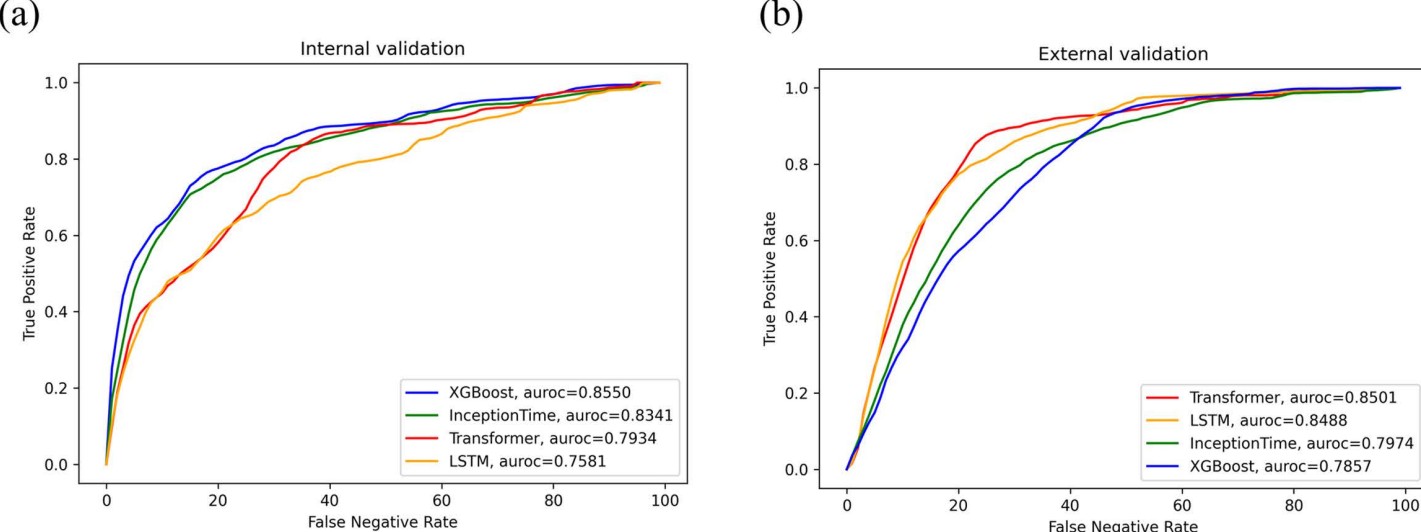

(a) Internal validation

(b) External validation

**Fig 3. Comparison of the performance curves of four models (XGBoost, InceptionTime, Transformer, and LSTM) in predicting intraoperative hypoxemia in pediatric patients under general anesthesia.** Abbreviations: AUROC for internal (a) and external validations **(b)**. LSTM, long short-term memory; AUROC, area under the receiver operating characteristic curve.

XGBoost on the internal validation dataset showed a decrease in AUROC from 0.855 to 0.804 and in and AUPRC from 0.182 to 0.139. For the external validation dataset, the performance of the Transformer decreased in AUROC from 0.850 to 0.711 and in AUPRC from 0.060 to 0.041. Second, the dataset was stratified into age subgroups for training. However, the model trained on the 2–8-year age subgroup showed performance improvements on the external validation dataset. This improvement was not consistent across all subgroups. As a result, these adjustments resulted in decreased overall performance (S4 and S5 Tables).

We also aimed to identify the minimal observation window required to effectively predict hypoxemia without significantly compromising performance. Therefore, we reduced the observation window from 1 minute down to 10 seconds and examined the resulting changes in performance. For XGBoost, increasing the observation window led to a slight improvement in AUROC of the internal validation dataset, rising from 0.836 to 0.855. However, the AUPRC increased significantly from 0.144 to 0.182. This indicated that the AUROC remained substantial even with a reduced 10-second observation window. With the increase in the observation window, the AUROC showed only a modest improvement, whereas a more substantial enhancement was observed in the AUPRC (S6 Table).

Additionally, we investigated the use of waveform biosignals for model performance enhancement. Three waveforms—photoplethysmography, airway pressure, and capnography—were converted into 2D spectrograms, and features were extracted using EfficientNet. These waveform features were concatenated with traditional single-measurement features derived using InceptionTime to generate the final prediction. The pretrained model achieved an AUROC of 0.8073 and an AUPRC of 0.0936 on the internal validation dataset and an AUROC of 0.7523 and an AUPRC of 0.0497 on the external validation dataset. The F1 scores were 0.0217 for internal validation and 0.0318 for external validation (S7 Table).

Finally, to address the potential for circular reasoning from using $SpO_2$ to predict a $SpO_2$-defined event, we evaluated the performance of models trained using only $SpO_2$ time-series data. The results, detailed in S8 Table, showed that the $SpO_2$-only models had lower performance across most metrics compared to our proposed multi-variable models. For instance, the AUROC of the multi-variable XGBoost model on the internal validation set was 0.855, significantly higher than the 0.766 achieved by its $SpO_2$-only counterpart. This demonstrates that incorporating additional physiological

variables provides critical contextual information that enhances the model's predictive power beyond simple SpO$_2$ trend extrapolation.

## Feature importance analysis

To visually explain the impact of the variables, we used SHAP to illustrate how they influence the model's predictions (Fig 4). The demographic information shows wide distribution of SHAP values across both positive and negative domains suggests their influence is highly context-dependent, varying based on interactions with other physiological variables. The SHAP values for SpO$_2$, FiO$_2$, and EtCO$_2$ show a broader distribution with greater density of points indicating both positive and negative effects on the model's predictions, compared to those for PIP, TV, and MV. This indicates that SpO$_2$, FiO$_2$, and EtCO$_2$ have a more significant impact on the model's predictions. Additionally, the variables PIP, TV, and MV predominantly contributed negatively to the model's predictions, as their distributions are primarily located on the left side of zero. This suggests that higher values of these variables may adversely affect the outcome.

## Discussion

This study developed various machine learning models to predict intraoperative hypoxemia in pediatric patients undergoing general anesthesia and performed external validation. The results showed that the XGBoost model demonstrated the best performance in internal validation, whereas the Transformer model outperformed others in external validation. Various methods were applied to enhance the performance of the hypoxemia prediction models, revealing that normalization of input values or stratification into age subgroups had minimal impact on performance. Additionally, reducing the observation window from 1 minute to 10 seconds maintained a high AUROC but resulted in a decreased AUPRC. Notably, SpO$_2$, FiO$_2$, and EtCO$_2$ were identified as having a greater impact on model predictions compared to PIP, TV, and MV.

Pediatric anesthesia poses unique risks because of narrow safety margins, limited drug choices, and the uneven distribution of specialized anesthesiologists [22]. Therefore, in medical environments where it is difficult to allocate dedicated pediatric anesthesiologists, machine learning-based predictive models could potentially serve as a supportive tool to help

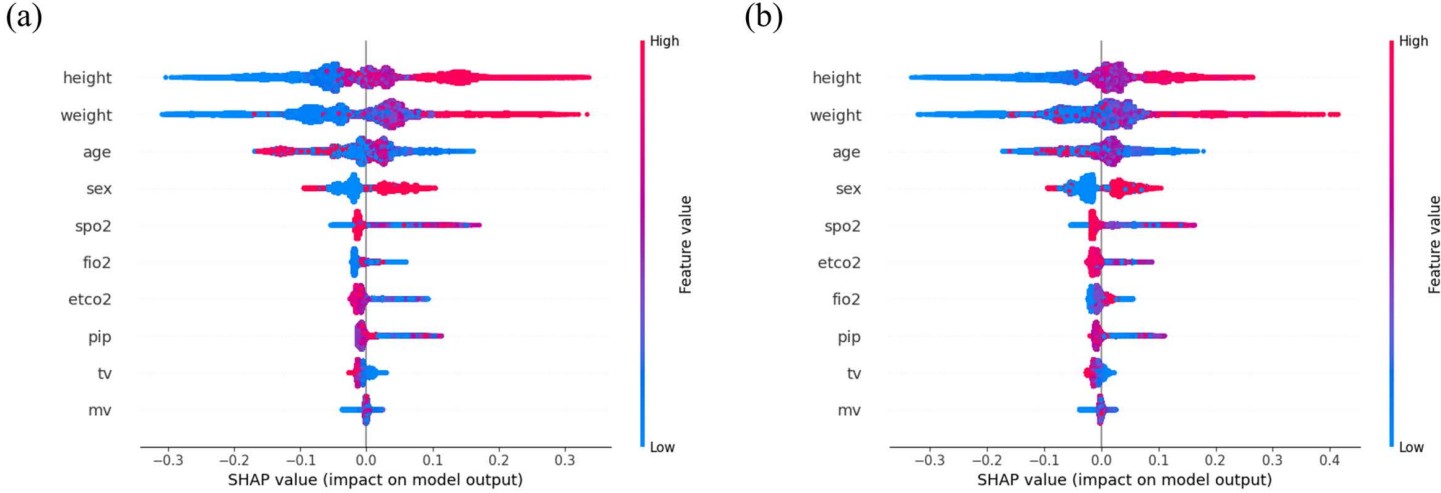

**Fig 4. SHAP values for feature importance in the model's predictions of intraoperative hypoxemia.** SHAP values for internal (a) and external validations **(b)**. The plot shows that demographic features have a highly context-dependent influence. In contrast, key physiological variables such as SpO$_2$, FiO$_2$, and EtCO$_2$ exhibit a more significant and direct impact on the model's predictions compared to mechanical ventilation parameters like PIP, TV, and MV, aligning with clinical intuition. Abbreviation: SHAP, Shapley Additive exPlanations; SpO$_2$, peripheral oxygen saturation; FiO$_2$, fraction of inspired oxygen; EtCO$_2$, end-tidal carbon dioxide; PIP, peak inspiratory pressure; TV, tidal volume and MV, minute ventilation.

address this gap. Previous studies on hypoxemia prediction have demonstrated that deep learning models perform as well as or better than traditional machine learning models [23]. However, other research has shown that machine learning approaches, such as the GBM, achieved the best performance for hypoxemia prediction [13]. Our findings align with this, indicating that even machine learning models like XGBoost, which lack a native sequential architecture [24], can achieve high performance by effectively capturing the complex interactions between biosignals and demographic data. Furthermore, as with previous studies, it is crucial to evaluate models using external validation rather than relying solely on internal validation to confirm their generalizability [10–13].

In cases such as inadequate ventilation, abnormal values may be observed in ventilator-related parameters such as $EtCO_2$, PIP, TV, and MV. TV and MV vary substantially with age, necessitating subgroup analysis and min-max normalization. These adjustments did not markedly improve performance, and SHAP values indicated relatively modest contributions from TV and MV, which were offset by integrating demographic data into the model's final layer.

Decreasing the observation window from 1 minute to 10 seconds revealed that even with a mere 10-second observation window, the AUROC remained substantially unchanged. This implies that the model primarily predicts hypoxemia using the absolute values just before a hypoxemic event rather than by analyzing trends within the time-series data. This finding clarifies why models like GBM can excel in time-series data. Particularly, the $SpO_2$ value, a key parameter used in training, serves both as a pivotal criterion for determining hypoxemic events and is used in prediction as well. Prior research on hypoxia prediction indicates that models relying solely on prior $SpO_2$ data as a variable generally outperform those incorporating multiple clinical variables [11]. From the SHAP value, $SpO_2$ significantly influences the model's predictions more than other variables.

Additionally, AUPRC values tended to increase as the observation window length increased. AUPRC is more effective for evaluating classifier performance in highly imbalanced datasets [25]. Although a high AUROC is beneficial, a clinically valuable model should also prioritize generating true-positives, illustrated by higher F1 scores and AUPRC values. As roughly 54% of actual hypoxemic events are accompanied by low $SpO_2$, refining predictive accuracy for early detection remains a priority.

Lastly, we aimed to improve the prediction of pediatric hypoxemia by incorporating photoplethysmography, airway pressure, and capnography waveforms, but it did not significantly enhance the model's performance and, in some cases, added noise. This outcome underscores the complexity of effectively integrating waveform features from multiple sources. Careful feature selection and tailored model design are essential to optimize predictions for pediatric hypoxemia.

This study has several limitations. The study is limited by biases inherent in its retrospective analysis, insufficient evidence linking hypoxemia prediction in children to reduced event occurrences, and the exclusion of high-risk groups, which restricts the applicability of the results. Furthermore, while our models demonstrated predictive capability significantly better than baseline, their precision, as measured by AUPRC, remains modest. This indicates a high rate of false alarms, which could lead to alarm fatigue. Notably, this discrepancy between high AUROC and modest AUPRC is consistent with previous findings on a similar high-frequency pediatric dataset [13], suggesting it may be an inherent challenge of this specific prediction task. However, this reflects a necessary trade-off in a clinical context where failing to predict a true hypoxemic event (a false negative) carries significantly greater risk than a false alarm. This trade-off solidifies the model's current role as a situational awareness support tool rather than a definitive diagnostic alarm. The model primarily focuses on ventilation-related measures, overlooking circulatory factors that also contribute to hypoxemia. Incorporating variables such as blood pressure and ECG data could improve the model's relevance to real-world surgical scenarios. Future research should adopt a prospective approach with a broader patient demographic, include real-time data collection to enhance precision and clinical applicability, and prioritize external validation across multiple institutions to ensure the model's generalizability and robustness.

In conclusion, we used machine learning models, specifically XGBoost and Transformer, to predict intraoperative hypoxemia in pediatric patients under general anesthesia. Internal and external validations were conducted using data

from two distinct hospitals (CHUH and SNUH), demonstrating robust predictive performance. It is expected that the proposed model, with improved precision and effectiveness and validated generalizability through external testing at additional institutions, has the potential for widespread adoption in clinical practice to accurately identify hypoxemia in pediatric patients.

## Supporting information

**S1 Table. TRIPOD-AI checklist.**
(DOCX)

**S2 Table. Counts of each labeled segment in the dataset.** This table presents the distribution of labeled segments across the training, internal validation, and external validation datasets, highlighting that hypoxemia-labeled segments were more prevalent in the CNUH dataset (1.42%) than in the SNUH datasets (1.10% in training and 1.08% in internal validation). Abbreviations: SNUH, Seoul National University Hospital; CNUH, Chungnam National University Hospital.
(DOCX)

**S3 Table. Hyperparameter search space and final selected values for the machine learning models.**
(DOCX)

**S4 Table. Comparative performance of machine learning models for hypoxemia prediction in pediatric patients after normalization.** The performance of the XGBoost and Transformer models for predicting hypoxemia in pediatric patients under general anesthesia was compared before and after normalization. The performance metrics, including the AUROC, AUPRC, and F1 score, were evaluated on both internal and external validation datasets. The highest values for each metric in both datasets are highlighted in bold. Abbreviations: AUROC, area under the receiver operating characteristic curve; AUPRC, area under the precision-recall curve; w/o, without; w/, with; norm., normalization.
(DOCX)

**S5 Table. Comparative performance of machine learning models for hypoxemia prediction in pediatric patients after training by age subgroup.** The performance of the XGBoost and Transformer models for hypoxemia prediction in pediatric patients under general anesthesia, stratified by age subgroups, shows variations in the AUROC, AUPRC, and F1 scores across internal and external validation datasets. Abbreviations: AUROC, area under the receiver operating characteristic curve; AUPRC, area under the precision-recall curve.
(DOCX)

**S6 Table. Comparative performance of the XGBoost model for hypoxemia prediction in pediatric patients based on changes in observation window length.** This table presents the performance of the XGBoost model for hypoxemia prediction in pediatric patients under general anesthesia, highlighting how different observation window lengths (from 10 to 60 seconds) impact the AUROC, AUPRC, and F1 scores across internal and external validation datasets. Abbreviations: AUROC, area under the receiver operating characteristic curve; AUPRC, area under the precision-recall curve.
(DOCX)

**S7 Table. Comparative performance of the machine learning model for hypoxemia prediction in pediatric patients using waveform biosignals.** In this analysis, the three waveforms—Photoplethysmography, Airway Pressure, and Capnography—were converted into 2D spectrograms, and features were extracted using EfficientNet. These waveform features were then concatenated with features derived from traditional single measurements, which had been extracted using InceptionTime, to be used for the final prediction. Abbreviations: AUROC, area under the receiver operating characteristic curve; AUPRC, area under the precision-recall curve.
(DOCX)

**S8 Table. Comparative performance of the machine learning model for hypoxemia prediction in pediatric patients using only the SpO$_2$ feature.** This table presents the performance of the four models when trained and evaluated using only the time-series of peripheral oxygen saturation (SpO$_2$) as an input feature. These results serve as a baseline to assess the added predictive value of the multi-variable model. Abbreviations: LSTM, long short-term memory; AUROC, area under the receiver operating characteristic curve; AUPRC, area under the precision-recall curve.
(DOCX)

## Acknowledgments

The IRBs of CHUH and SNUH waived the requirement for informed consent owing to the retrospective nature of the study.

## Author contributions

**Conceptualization:** Jung-Bin Park, Kyungsang Kim, Dongheon Lee, Boohwi Hong.

**Data curation:** Jung-Bin Park, Hyung-Chul Lee, Boohwi Hong.

**Funding acquisition:** Boohwi Hong.

**Resources:** Sujin Baek, Boohwi Hong.

**Software:** Ji Hye Heo, Donghyeon Baek, Dongheon Lee.

**Supervision:** Kyungsang Kim, Chahyun Oh, Hyung-Chul Lee, Dongheon Lee, Boohwi Hong.

**Validation:** Sujin Baek, Jung-Bin Park, Donghyeon Baek.

**Visualization:** Ji Hye Heo, Donghyeon Baek.

**Writing – original draft:** Sujin Baek, Ji Hye Heo.

**Writing – review & editing:** Kyungsang Kim, Chahyun Oh, Dongheon Lee, Boohwi Hong.

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
