## [Decision Letter · Decision Letter 0]

15 Aug 2025

Dear Dr. Hong,

Thank you for submitting your manuscript to PLOS ONE. After careful consideration, we feel that it has merit but does not fully meet PLOS ONE’s publication criteria as it currently stands. Therefore, we invite you to submit a revised version of the manuscript that addresses the points raised during the review process.

We look forward to receiving your revised manuscript.

Kind regards,

Vijayalakshmi Kakulapati, Ph.D

Academic Editor

PLOS ONE

Journal Requirements:

4. Thank you for stating the following in your manuscript:

“This research was supported and funded by the SNUH Lee Kun-hee Child Cancer & Rare Disease Project, Republic of Korea (grant number: 24C-003-0100) and supported by research fund from Chungnam National University.”

6. We notice that your supplementary [figures/tables] are included in the manuscript file. Please remove them and upload them with the file type 'Supporting Information'. Please ensure that each Supporting Information file has a legend listed in the manuscript after the references list.

Additional Editor Comments (if provided):

Reviewers' comments:

Reviewer's Responses to Questions

**Comments to the Author**

1. Is the manuscript technically sound, and do the data support the conclusions?

Reviewer #1: Yes

Reviewer #2: Yes

2. Has the statistical analysis been performed appropriately and rigorously?

Reviewer #1: Yes

Reviewer #2: I Don't Know

3. Have the authors made all data underlying the findings in their manuscript fully available?

Reviewer #1: Yes

Reviewer #2: No

4. Is the manuscript presented in an intelligible fashion and written in standard English?

Reviewer #1: Yes

Reviewer #2: Yes

Reviewer #1: This study addresses pediatric hypoxemia prediction using machine learning models. While technically sound, several methodological issues must be clear up before publication.

Major Concerns:

1. Circular Logic in SpO2-based Prediction

The study defines hypoxemia as SpO2 < 95% while simultaneously using SpO2 as the primary predictor variable. SHAP analysis confirms SpO2 as the most important feature, creating a circular prediction framework. To demonstrate true clinical value, the authors should compare performance against: (1) SpO2-only models, (2) simple threshold-based rules, and (3) existing clinical practice standards if possible. Or state it on limitation.

2. Poor Precision Performance

The AUPRC values are critically low: XGBoost achieved 0.1816 (internal) and 0.0402 (external), while LSTM achieved 0.0605(external). The F1 scores remain poor (XGBoost: 0.2382 internal, 0.0824 external). These metrics indicate severely limited precision in clinical application, yet the authors fail to adequately discuss whether such performance justifies implementation or potential harm from false alarms.

3. Inadequate Prediction Timeline Justification.

The authors should justify why they defined a 1-minute prediction window. Is this timeframe sufficient for clinical intervention or prevention?

Reviewer #2: Many thanks for providing the manuscript for review. The authors utilise physiological data captured from intra-operative anaesthesia devices in order to predict the onset of hypoxemia in a subset of paediatric patients undergoing general anaesthesia in the Republic of Korea. They utilise a rolling window method in which 120 second segments of data are classified as either hypoxemia or no hypoxemia and make use of the first half (60 seconds) to predict occurrence of any hypoxemia in the second (61-120 seconds) half of the window.

The models utilise include XGBoost, LSTM, transformer model and a CNN approach (InceptionTime). The authors report a high AUROC performance for each of the models and utilise this to perform external validation on patients from another institution.

Comments

1. I would like to see the Tripod AI checklist included within the supplementary material provided

2. It would be useful to have further clarity in the introduction on the relationship between a low SpO2 reading and true hypoxemia – what criteria is used to define hypoxemia conventionally e.g. will this include low arterial blood gas oxygen pressures? Helpful for readers who do not work within anaesthesia for example.

3. I would like to also see data on the different types of surgery cases included in the baseline descriptions to understand the population better.

4. The authors mention that relying on SpO2 readings alone is insufficient for accurately assessing true hypoxemia – however their definition of hypoxemia used for model training and evaluation appears to be based largely on SpO2 alone. Can they elaborate on what processes are used to ensure this is clinically valid (e.g. annotation process)

5. What is the nature of the ‘vital signs’ used to train the initial models? Are these tabular data extracted from the machines, continuous waveforms or a composite – useful to be explicit and describe the data, data quality, and what pre-processing work was involved in making them useable.

6. How were the hyperparameters chosen? Is this through cross-validation if so what are the performances? Reporting only the hold out set metrics for model development may not provide the full picture.

7. The model utilises SpO2 as their primary input feature in order to predict subsequent SpO2. How is the nature of repeated measurement accounted for in the models (if they need to be). For example, I would imagine a low SpO2 in the first half of the window is predictive of a low SpO2 in the second half?

8. Can the authors comment on why XGBoost was the best performing in the internal validation set but did not perform as well in external validation? There is a disproportionate difference between XGBoost and LSTM approaches internally.

9. The dataset is highly imbalanced with only around 1.4% at most positive. This means that the AUROC will be high regardless and therefore can be misleading to present this as the primary result. The AUPRC for the external validation is around 0.06 (where the baseline will be around 0.012) – can the authors comment on the suitability of the model for clinical implementation? Is this a useful model to inform clinicians?

10. I would disagree with the statement that machine learning models are not inherently designed (or suitable) for time series data – when alternatives include linear modelling.

11 . What is the significance of the SHAP analysis and figure – especially when height is the greatest contributor to model prediction but most features are centred around 0.

12. Could the captions and figures be provided in higher resolution?

13. Can I urge the anonymised data to be provided as part of submission given this would be valuable contribution to medical research into signal analysis where possible.

Many thanks

**Do you want your identity to be public for this peer review?** For information about this choice, including consent withdrawal, please see our Privacy Policy

Reviewer #1: No

Reviewer #2: **Yes: ** Damien Ming

---

## [Author Response · Author response to Decision Letter 1]

22 Sep 2025

We have uploaded our detailed responses to the reviewers’ comments as a separate file. We sincerely appreciate the reviewers’ thoughtful feedback. We have carefully checked both the manuscript and the submission system, and we confirm that the funding information has been provided consistently in both.

---

## [Decision Letter · Decision Letter 1]

4 Dec 2025

Hypoxemia prediction in pediatric patients under general anesthesia using machine learning: a retrospective observational study and external validation

PONE-D-25-28107R1

Dear Dr. Hong,

We’re pleased to inform you that your manuscript has been judged scientifically suitable for publication and will be formally accepted for publication once it meets all outstanding technical requirements.

Kind regards,

Vijayalakshmi Kakulapati, Ph.D

Academic Editor

PLOS ONE

Reviewers' comments:

Reviewer #1: All comments have been addressed

Reviewer #3: All comments have been addressed

Reviewer #1: The author's revised manuscript has successfully addressed all scientific concerns raised during the review process. The revisions demonstrate a thorough understanding all comments and provide satisfactory responses to each point. The scientific rigor, methodology, and interpretation of results have been appropriately strengthened in accordance with the feedback provided.

I suggested this revised version justifying for publication.

---

## [Editor Report · Acceptance letter]

PONE-D-25-28107R1

PLOS One

Dear Dr. Hong,

I'm pleased to inform you that your manuscript has been deemed suitable for publication in PLOS One. Congratulations! Your manuscript is now being handed over to our production team.

Kind regards,

on behalf of

Dr. Vijayalakshmi Kakulapati

Academic Editor

PLOS One